# Brimonidine Modulates the ROCK1 Signaling Effects on Adipogenic Differentiation in 2D and 3D 3T3-L1 Cells

**DOI:** 10.3390/bioengineering9070327

**Published:** 2022-07-19

**Authors:** Araya Umetsu, Yosuke Ida, Tatsuya Sato, Megumi Watanabe, Yuri Tsugeno, Masato Furuhashi, Fumihito Hikage, Hiroshi Ohguro

**Affiliations:** 1Department of Ophthalmology, School of Medicine, Sapporo Medical University, Sapporo 060-8556, Japan; araya.umetsu@sapmed.ac.jp (A.U.); y.ida@sapmed.ac.jp (Y.I.); watanabe@sapmed.ac.jp (M.W.); yuri.tsugeno@gmail.com (Y.T.); fuhika@gmail.com (F.H.); 2Department of Cardiovascular, Renal and Metabolic Medicine, Sapporo Medical University, Sapporo 060-8556, Japan; satatsu.bear@gmail.com (T.S.); furuhasi@sapmed.ac.jp (M.F.); 3Department of Cellular Physiology and Signal Transduction, Sapporo Medical University, Sapporo 060-8556, Japan

**Keywords:** deepening of the upper eyelid sulcus (DUES), 3T3-L1cells, ROCK, Rho-kinase, ROCK inhibitor, 3-dimension (3D) tissue culture

## Abstract

The additive effects of an α2-adrenergic agonist, brimonidine (BRI), on the pan-ROCK inhibitor (ROCK-i), ripasudil (Rip), and the ROCK2-I, KD025, on adipogenic differentiation (DIF+) were examined using two- or three-dimension (2D or 3D) cultures of 3T3-L1 cells. The following analyses were carried out: (1) lipid staining (2D and 3D), (2) real-time measurements of cellular metabolism (2D), (3) mRNA expression of DIF+ related genes and extracellular matrix molecules (ECMs) including collagen (Col)-1, -4, and -6, and fibronectin (Fn), and (4) the sizes and physical properties of the 3D spheroids. The findings indicate that DIF+ induced (1) a substantial enhancement in lipid staining and enhanced expression of the Pparγ and Fabp4 genes, (2) significantly larger and softer 3D spheroids, and (3) down-regulation of Col1 and Fn and up-regulation of Col4 and Col6 genes. Treatment with Rip alone caused a significant enhancement in adipogenesis of both the 2D and 3D cultured 3T3-L1 cells and in the physical properties of the 3D spheroids; these effects were substantially inhibited by BRI, and the effects induced by BRI or KD025 were not insignificant. These collective findings indicate that the addition of BRI inhibited the Rip-induced enhancement of DIF+ in 3T3-L1 cells, presumably by modulating ROCK1 signaling.

## 1. Introduction

Prostaglandin analogues (PGs) are currently used as a first-line anti-glaucoma medication because of their fewer side effects and their remarkable hypotensive effects [1,2,3,4]. However, in cases where the hypotensive effects of these analogs are insufficient for the treatment of glaucomatous optic neuropathy (GON), several other therapeutic options are used. These include additional anti-glaucoma medications, including β-blockers, α-agonists, carbonic anhydrase inhibitors, and a Rho-associated coiled-coil-containing protein kinase (ROCK) inhibitor (ROCK-i), as well as surgical intervention [5]. To select additional anti-glaucoma medications in addition to the first-line PGs, individual GON conditions, as well as the systemic and local side effects of each medication, need to be considered.

It should be noted, however, that local side effects caused by PGs have been recently reported. These include prostaglandin-associated periorbitopathy (PAP) such as the deepening of the upper eyelid sulcus (DUES) (presumably caused by orbital fat atrophy), and this has attracted the attention of clinicians [6,7,8,9,10]. In fact, such fat atrophy induced by PGs was indeed confirmed by previous in vitro studies using two-dimensional (2D) cultures of 3T3-L1 cells, which belong to the most popular pre-adipocyte cell line used in the field of lipid research [11]. Our group also established a more suitable in vitro model using a three-dimensional (3D) spheroid culture that replicates the spreading of adipocytes within the orbital cone environment [12], and found that PGs induced significant alterations in the expression of extracellular matrix (ECM) molecules in addition to adipogenesis in 3T3-L1 cells [13], as well as human orbital fibroblasts (HOFs) [14].

Adipocytes have various pivotal functions and properties including lipid storage, energy homeostasis, and insulin sensitivity, and their adipogenic differentiation (DIF+) is critically regulated by transcriptional factors including peroxisome proliferator-activated receptor γ (PPARγ), as well as others [15], and are influenced by several signaling molecules including PGs [16], ROCK, and others [17]. Recently, we also investigated the effects of ROCK inhibitors (ROCK-is) on DIF+ using 2D- and 3D-cultured 3T3-L1 cells, since ROCK-is are known to reduce the deposits of ECM proteins within the trabecular meshwork (TM), resulting in IOP reduction in several animal models [18,19]. One pan-ROCK-i, ripasudil hydrochloride hydrate (Rip), is already approved for the treatment of glaucoma and ocular hypertension [20,21]. Interestingly, our results revealed that the addition of ROCK-i beneficially abrogated the PG-induced suppression of DIF+ in 3T3-L1 cells, suggesting that ROCK-i may be one of the preferred choices as a second-line glaucoma medication for reducing the DUES that is induced by PGs [22].

Quite recently, as an additional option for anti-glaucoma medication for use in treating patients with advanced GON, a new fixed combination of Rip and an α2-adrenergic agonist, brimonidine (BRI), will soon become available in our glaucoma clinic. This substance takes BRI-induced neuroprotective effects into account, in addition to their synergistic hypotensive effects. However, as of this writing, no information is currently available regarding the effects of such a fixed combination of Rip and BRI on orbital fatty tissues. Furthermore, pharmacologically, the means by which ROCK1-i or ROCK2-i action is primarily involved in Rip-induced effects is also unidentified, since it is well-known that the two isoforms, ROCK1 and ROCK2, are involved in the pathophysiology of several organs, including the eye, in different manners [23,24].

Therefore, in the current study, we evaluated the additive effects of BRI to ROCK-is, pan-ROCK-i, Rip, and ROCK2-i KD025 on DIF+ of 2D- and 3D-cultured 3T3-L1 cells based on DIF+ (2D and 3D cultures), real-time cellular metabolism measurements (2D), ECM expression (2D and 3D), and physical properties, size, and stiffness (3D).

## 2. Materials and Methods

### 2.1. 2D and 3D Cultures and DIF+ (Adipogenic Differentiation) of 3T3-L1 Cells

The 2D and 3D cultures and adipogenic differentiation (DIF+) of 3T3-L1 preadipocytes (DIF-) (#EC86052701-G0, KAK) were processed as described previously [12,22,25]. Briefly, 2D 3T3-L1 cells were maintained at 37 °C in a 2D culture medium composed of HG-DMEM supplemented with 8 mg/L d-biotin, 4 mg/L calcium pantothenate, 100 U/mL penicillin, 100 μg/mL streptomycin, and 10% CS. For further 3D spheroid culture, approximately 20,000 cells in 28 μL of 3D spheroid medium composed of the 2D culture medium supplemented with 0.25% *w*/*v* Methocel A4M were cultured in each well of a drop culture plate and maintained by replacing half of the culture medium daily until day 7.

Adipogenic differentiation (DIF+) of 2D or 3D 3T3-L1 cells was induced by supplementation with 250 nM dexamethasone, 10 nM T3, 10 μM troglitazone, and 1 μg/mL insulin during days 1–2, and then with 10 μM troglitazone and 1 μg/mL insulin.

To estimate the drug-induced effects of the ROCK-is, 10 µM Ripasudil (Rip, a generous gift from the Kowa Company Ltd., Nagoya, Japan) or KD025 (Sigma-Aldrich, St Louis, MO, USA), and/or 10 mM brimonidine (BRI) were added during days 1–7. These drug concentrations were confirmed as the optimum concentrations based on our previous studies in which ROCK-is was used [22] and a cell viability analysis of BRI using retinal pigment epithelium cells [25].

### 2.2. Measurement of Real-Time Cellular Metabolic Functions

The oxygen consumption rate (OCR) and extracellular acidification rate (ECAR) of the 2D-cultured 3T3-L1 preadipocyte (DIF-) and those with adipogenic differentiation (DIF+) in the absence or presence of 10 µM ROCK-i, Rip or KD025, and/or BRI were measured using a Seahorse XFe96 Bioanalyzer (Agilent Technologies, Santa Clara, CA, USA) according to the manufacturer’s instructions. In a typical analysis, 20 × 10^3^ 2D cultured cells were placed in the well of a XFe96 Cell Culture Microplate (Agilent Technologies, #103794-100). After centrifuging the plate at 1600× *g* for 10 min, the culture medium was replaced with 180 μL of assay buffer (Seahorse XF DMEM assay medium (pH 7.4, Agilent Technologies, #103575-100), supplemented with 5.5 mM glucose, 2.0 mM glutamine, and 1.0 mM sodium pyruvate). For the 3D-cultured cells, the spheroids were washed twice with phosphate-buffered saline and five individual spheroids were then transferred to the well of a XFe96 Spheroid Microplate (Agilent Technologies, # 102978-100) containing 180 μL of the assay buffer. The assay plates were incubated in a CO2-free incubator at 37 °C for 1 hour prior to the measurements. OCR and ECAR were measured in the Seahorse XFe96 Bioanalyzer under a 3 min mix and a 3 min measure protocol at the baseline and subsequent injections of oligomycin (final concentration: 2.0 μM), carbonyl cyanide p-trifluoromethoxyphenylhydrazone (FCCP, final concentration: 5.0 μM), a mixture of rotenone/antimycin A (final concentration: 1.0 μM), and 2-deoxyglucose (2-DG, final concentration: 10 mM).

### 2.3. Lipid Staining of the 2D and 3D 3T3-L1 Cells

Lipid in the 2D and 3D 3T3-L1 cells was stained with Oil Red O lipid stain and a BODIPY lipid stain, respectively, as described previously [22]. For quantification of the lipid staining levels, the optical density (O.D.) of the extracted Oil Red O dye and the fluorescence intensity of the BODIPY-stained lipid droplets were measured [22].

### 2.4. Physical Property (Size and Stiffness) Measurements of the 3D 3T3-L1 Spheroids

Measurements of the physical properties, size, and stiffness of the 3D 3T3-L1 spheroids were performed by analyzing phase-contrast microscopy images using the Image J software and a micro-indentation analysis using a micro-squeezer (CellScale, Waterloo, ON, Canada), as described previously [26]. As an index, the force/displacement (μN/μm) for the stiffness of the 3D spheroid, defined as the force required (μN) to achieve a 50% deformation over 20 s, was measured.

### 2.5. Immunocytechemictry of 3T3-L1 Cells

As described in our previous studies [14,27], 4% paraformaldehyde-fixed 3T3-L1 cells (2D and 3D) were successively incubated with 1:200 dilutions of 1st antibodies; an anti-human COL1, COL4, COL6, FN, or αSMA rabbit antibody and 1:1000 dilutions of 2nd antibody; a goat anti-rabbit IgG (488 nm) with 1:1000 dilutions of phalloidin; and DAPI. Thereafter, their confocal immunofluorescent images were obtained.

### 2.6. Other Analytical Methods

The RNA extraction, the reverse transcription, and the following real-time PCR were processed as described previously [28], using specific predesigned primers (Integrated DNA Technologies, Inc., Coralville, IA, USA), as shown in Appendix A. All statistical analyses were performed using Graph Pad Prism 8 (GraphPad Software, San Diego, CA, USA), as described previously [29].

## 3. Results

To study the additive effect of BRI to ROCK-is on orbital fatty tissues in the case where a fixed combination of BRI and Rip was used, we evaluated the additive effects of BRI to ROCK-i (Rip or KD025) on several properties of 2D- and 3D-cultured 3T3-L1 cells, including lipid staining (2D and 3D), real-time measurements of cellular metabolism (2D), the mRNA expressions of DIF+ related genes and ECM proteins, and the physical properties, namely, the size and stiffness of the 3D spheroids.

As shown in Figure 1, lipid staining with Oil Red O and the quantitative PCR of Pparγ and Fabp4 for the 2D cultured 3T3-L1 cells were increased upon adipogenic differentiation (DIF+), as described in our own, and other, previous studies [13]. The addition of pan-ROCK-i, Rip, or ROCK2-i KD025 to DIF+ 2D cultured 3T3-L1 cells caused significant enhancement in the lipid staining by the Oil Red O and a significant suppression of the mRNA expressions of Pparγ and Fabp4, respectively. In contrast, while a mono-treatment of BRI did not induce significant effects in the lipid-staining and these DIF+ related genes (except for the down-regulation of Pparγ expression), BRI significantly suppressed the Rip-enhanced effects of DIF+, as described above. In terms of the mRNA expression of ECM molecules in the 2D-cultured 3T3-L1 cells, down-regulation was observed in the levels of Col and up-regulation was observed in the levels of COL6 upon DIF+, while ROCK-is and/or BRI exerted a significant effect on the Col1 (Bri, up-regulation; Rip, down-regulation), Col4 (Rip, up-regulation), and Col6 (KD, down-regulation; BRI/KD, down-regulation) of DIF+ expressions (Figure 2). These significant differences among the experimental conditions that were examined were also confirmed by immunocytochemistry (Appendix A). Measurement of real-time cellular metabolism by a Seahorse analyzer indicated that (1) there were no significant changes upon DIF+ in the 2D 3T3-L1 cells in the OCR in response to FCCP injection, which reflects mitochondrial maximal respiratory capacity (MMRC), or the ECAR in response to oligomycin injection, which reflects glycolytic reserve (GR); (2) MMRC was enhanced, not affected or suppressed, by Rip and/or BRI, KD025, or KD and BRI, respectively, and (3) GR was enhanced by Rip or BRI and KD025 and not affected by BRI, KD025, or BRI and Rip, respectively (Figure 3). These collective data indicate that DIF+ of the 2D-cultured 3T3-L1 cells were affected by Rip and KD025 in different manners and that most of the Rip-induced effects were inhibited by the addition of BRI, although the mitochondrial oxidative phosphorylation (OXPHOS) and glycolysis of these cultures were modulated differently among the mono-treatment and several combinations of ROCK-is and BRI.

We next studied the additive effect of BRI on ROCK-is, Rip, or KD025 in 3D 3T3-L1 spheroids, which have been established as a reliable in vitro model for DUES [13,14]. As shown in Figure 4 and consistent with findings reported in our previous study, upon DIF+, the 3D spheroids became significantly larger and softer, and these effects were further enhanced by Rip. A mono-treatment with BRI or KD025 had no significant effect on the DIF+-induced sizes and stiffness of the 3D spheroids. However, BRI substantially inhibited the Rip-induced enlargement and inhibited the softening effects, respectively. Therefore, these results suggest that Bri may suppress the ROCK1 inhibitory effects on the physical properties of the DIF+ 3D 3T3-L1 spheroids.

To further study the additive effects of BRI to ROCK-is on DIF+ in 3D 3T3-L1 spheroids, lipid staining with BODIPY and the mRNA expression of DIF+-related genes, including Pparγ and Fabp4, were examined. As shown in Figure 5A,B, adipogenic differentiation (DIF+) induced an increase in the intensity of BODIPY staining and Pparγ and Fabp4 expressions were further enhanced by the mono-treatment of Rip, as was consistently observed in our previous report [22]. A KD025 mono-treatment also enhanced DIF+-induced lipid staining and the expressions of DIF+-related genes, but those effects were not as robust compared to Rip. In contrast, while these effects were not observed in the case of BRI mono-treatment, the ROCK-is induced effects were inhibited upon addition of BRI. In terms of the expression of EM molecules by the 3D 3T3-L1 spheroids, similar to the results of the 2D cell culture experiment described above, the DIF+-induced down-regulation of Col1 and Fn was observed (Figure 6), but no significant effects by ROCK-is and/or BRI were observed, except for the up-regulation of Fn by BRI and KD025. These significant differences among the experimental conditions that were examined were also confirmed by immunocytochemistry (Appendix A). These collective data strongly suggest that BRI substantially modulates the ROCK-is induced effects on the 2D- and 3D-cultured 3T3-L1 cells despite the insignificant effects that were observed for the BRI mono-treatment.

## 4. Discussion

The adipogenic differentiation that is required for adipocyte maturation is recognized to be critically regulated by several key genes, including the peroxisome proliferator-activated receptor γ (PPARγ) and the CCAAT enhancer-binding protein α (C/EBPα), both essential regulators of DIF+, and, as a result, the expression of other metabolic genes (including those for the glucose transporter 4 (GLUT4), the fatty acid binding protein 4 (FABP4), leptin and others) is induced [29,30,31,32]. It is also well-known that this main DIF+ process is greatly affected by several extracellular factors, including insulin/IGF-1, TGFβ, FGF, prostaglandin derivatives (PGs), and others [33]. In fact, recent studies have reported that the PGs-induced suppression of DIF+ in orbital adipocytes, a process that is referred to as DUES, has attracted interest in the field of ophthalmology because PGs are frequently used as the first-line anti-glaucoma medication [34]. Following the discovery of PG-induced DUES, the issue of whether or not other drugs used as instillations could also affect DIF+ of orbital adipocytes has attracted considerable interest. To examine this issue, we developed a drug screening system using 3D spheroid cultures of 3T3-L1 cells and human orbital fibroblasts (HOFs) in addition to conventional 2D cell cultures [13,14,27], and found that pan-ROCK-is, Rip and Y27632, and EP2 agonists, omidenepag (OMD) were also significantly affected, but in different manners, in 3T3-L1 cells [22,35] and HOFs [36]. Therefore, these observations suggest that our newly developed methods for evaluating drug efficacies toward adipocytes may be a promising strategy for further studies with respect to pharmacological mechanisms. Therefore, in the current study, to elucidate the unidentified roles of ROCK2 and α2-adrenergic signaling in the DIF+ of 3T3-L1 cells, the drug-induced effects of several combinations of pan-ROCK-i, Rip, specific ROCK2-i, KD025, and the α2-adrenergic agonist, brimonidine/BRI were investigated. The results indicated that (1) a mono-treatment of Rip, but not BRI or KD025, caused a significant enhancement in DIF+ in both 2D- and 3D-cultured 3T3-L1 cells and in the physical properties of the 3D spheroids, and (2) BRI inhibited the Rip-induced enhancement of DIF+, presumably by modulating ROCK1 signaling.

In a previous study, we reported that Rho/ROCK signaling was a negative regulator of adipocyte differentiation [37,38], similar to PGs [11]. Alternatively, α2-adrenergic receptors are expressed in adipocytes at high levels and play an important role as an adrenergic regulator of lipolysis [39]. However, such adrenergic regulation of lipolysis is known to be exclusively species-dependent; that is, human fat cells express both lipolytic β- and antilipolytic α2-adrenergic receptors (α2-ARs) [40,41], whereas no functional α2-ARs are found in rat fat cells [39]. In fact, an antilipolytic α2-adrenergic effect is well-known within human adipocytes [42,43]. Regarding mouse adipocytes, it was revealed that the proliferation of 3T3F442A preadipocytes can be amplified by an α2-adrenergic agonist. Interestingly, the α2-adrenergic-dependent effects were totally abolished by lysophospholipase, phospholipase B, and it was reported that lysophosphatidic acid (LPA) could induce the spreading and proliferation of 3T3F442A preadipocytes. These collective observations suggest that the α2-adrenergic stimulation of adipocytes induces the extracellular release of LPA, resulting, in turn, in the regulation of preadipocyte growth [44]. Quite interestingly, among these PGs, α-adrenergic, and ROCK signaling mechanisms, it has also been shown that PGF2α-related signaling is closely linked to the Ga12-ROCK signaling pathway [45,46], and α1-adrenergic vasoconstriction is linked with S1P2–G12/13–ROCK-mediated signaling [47]. However, our previous study indicated that the inhibition of ROCKs by pan-ROCK-is, Rip, or Y27632 additively, but not synergistically, affected the effects that are induced by PGs on DIF+ [22,48], suggesting that both types of signaling may be independent in terms of the DIF+ in 3T3-L1 cells. In contrast, the findings reported in this study indicate that the α2-adrenergic stimulation by BRI may function as an inhibitory link to ROCK1 signaling mechanisms, as above.

However, there are limitations to this study. The following issues require consideration: first, the roles of ROCK1 in the process of DIF+ are not fully understood since, among the ROCKs, ROCK2, but not ROCK1, is recognized as being mainly responsible for anti-adipogenic activity toward 3T3-L1 cells [49]. Therefore, additional investigations with the objective of elucidating the roles of ROCK1 and 2 signaling within the DIF+, as well as their relationship with α2-adrenergic signaling, remain outstanding. Such studies will require the use of additional experimental groups such as Rip and KD025, as well as other pan-ROCK-is molecules with different suppressive efficacies with respect to ROCK1 and 2. Second, in the current study, we used 3T3-L1 cells, a mouse embryonic fibroblast cell line, to study the drug-induced effects toward orbital fatty tissues. However, these adipocytes may have different characteristics, since it is known that obesity is associated with an increase in general adipocytes but not orbital adipocytes [50], but in turn, orbital adipocytes but not general adipocytes are increased in patients with Graves’ disease [51]. Third, the drug-induced effects of the mRNA expressions of ECMs molecules were quite different between 2D and 3D cell cultures, despite the lipid staining and mRNA expression of DIF+-related genes being similar. As a possible explanation for this difference, we speculate that DIF+ may prefer a 3D spatial space rather than the 2D plane space. In fact, in a previous study, we identified the spontaneous adipogenic differentiation of the 3D 3T3-L1 preadipocytes without their induction, although such spontaneous DIF+ was not detected in the 2D-cultured cells. To identify possible mechanisms responsible for causing such a difference between 2D and 3D cell cultures, RNA sequence analyses were performed. The results suggested that STAT3 functions as the master regulator in forming the 3D spheroid architecture, and related signaling may induce these unique biological differences [52]. However, as of this writing, the relationship between DIF+ and STAT3-related signaling, as well as diversity in the nature of orbital and other systemic adipocytes, has not been fully investigated. Therefore, additional study using several specific inhibitors, SiRNA knockdown, and other strategies will be required to reveal the relationship between biological environments, e.g., 2D vs. 3D, and various biological signaling networks, using several sources of adipocytes, i.e., both orbital and other systemic adipocytes.

## Figures and Tables

**Figure 1 bioengineering-09-00327-f001:**
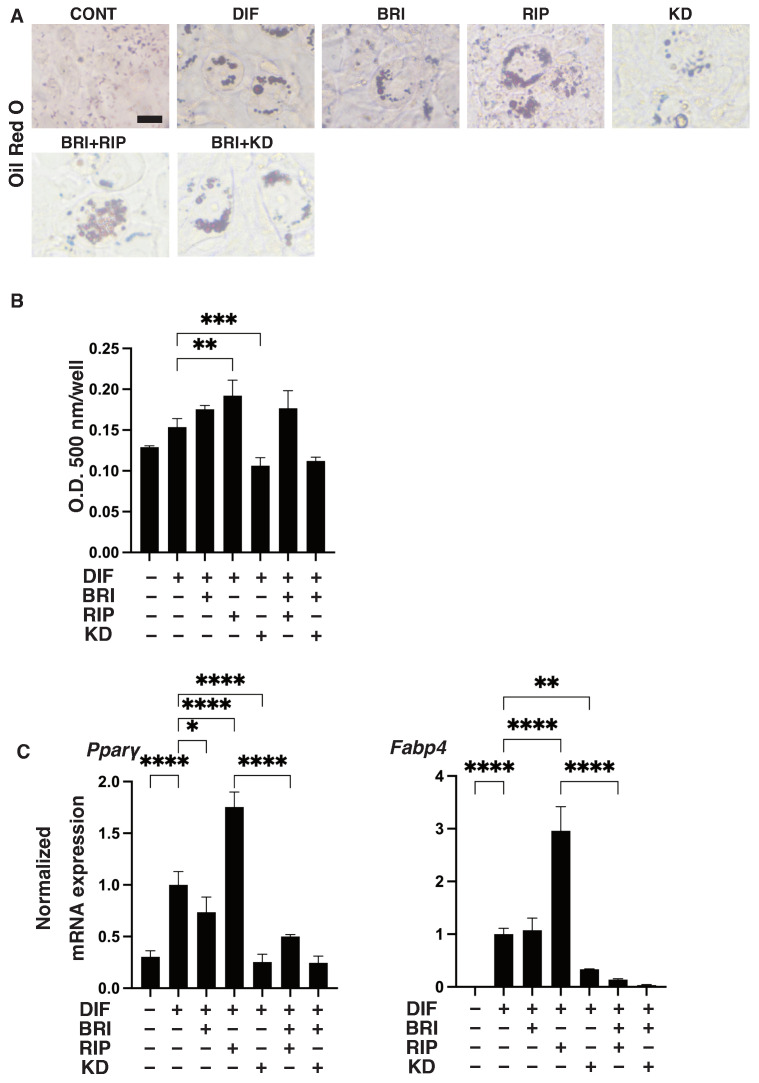
Effects of ROCK-is and/or BRI on adipogenic differentiation (DIF+) of 2D-cultured 3T3-L1 cells. The 3T3-L1 cells were 2D-cultured under several conditions: preadipocytes of 3T3-L1 cells (DIF−) and their adipogenic differentiation (DIF+) with or without 10 µM ripasudil (Rip) or KD025, and/or BRI. These were stained by Oil Red O (**A**, scale bar: 100 μm), and the staining intensities were plotted (**B**). The mRNA expressions of adipogenesis-related genes, including Pparγ and Fabp4, under the above conditions were plotted in (**C**) 1–3. All experiments were performed in triplicate using fresh preparations, each consisting of five samples. Data are presented as the arithmetic mean  ±  standard error of the mean (SEM). * *p* < 0.05, ** *p* < 0.01, *** *p* < 0.005, **** *p* < 0.001, and other pairs were not statistically significant (ANOVA followed by a Tukey’s multiple comparison test).

**Figure 2 bioengineering-09-00327-f002:**
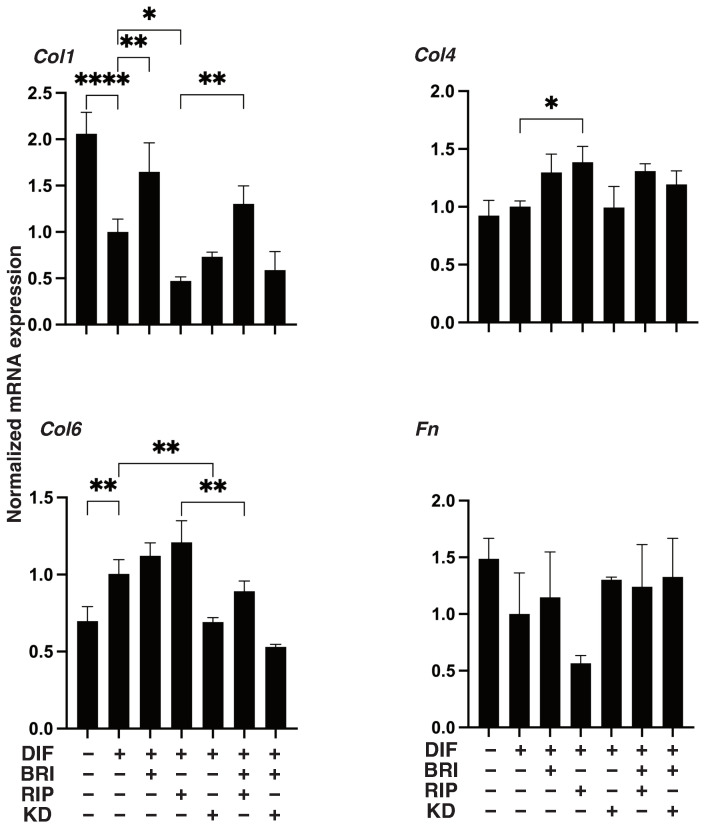
Effects of ROCK-is and/or BRI on ECM expression in 2D cultured 3T3-L1 cells. The 3T3-L1 cells were 2D-cultured under several conditions: preadipocytes of 3T3-L1 cells (DIF−) and their adipogenic differentiation (DIF+) with or without 10 µM ripasudil (Rip) or KD025, and/or BRI were subjected to qPCR analysis to estimate mRNA expression of ECM molecules (Col1: collagen 1, Col4: collagen 4, Col6: collagen 6, Fn: fibronectin). All experiments were performed in triplicate using fresh preparations, each consisting of five samples. Data are presented as the arithmetic mean  ±  standard error of the mean (SEM). * *p* < 0.05, ** *p* < 0.01, **** *p* < 0.001, and other pairs were not statistically significant (ANOVA followed by a Tukey’s multiple comparison test).

**Figure 3 bioengineering-09-00327-f003:**
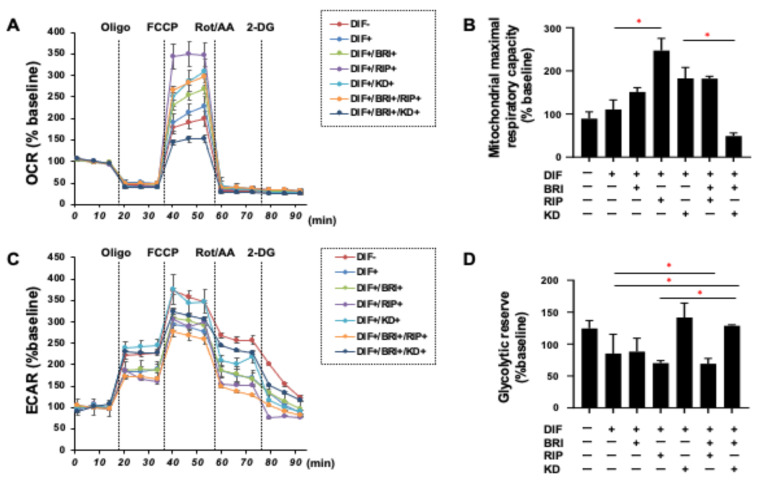
Real-time analysis of cellular metabolism of 2D-cultured 3T3-L1 cells. The 3T3-L1 cells were 2D-cultured under several conditions: preadipocytes of 3T3-L1 cells (DIF−) and their adipogenic differentiation (DIF+) with or without 10 µM ripasudil (Rip) or KD025, and/or BRI were subjected to a real-time metabolic function analysis using a Seahorse XFe96 Bioanalyzer. (**A**,**B**) Basal OCR and ECAR were measured, and thereafter they were further measured after subsequent supplementation with oligomycin (complex V inhibitor), FCCP (a protonphore), and rotenone/antimycin A (complex I/III inhibitors) and 2DG (hexokinase inhibitor). (**C**,**D**) Basal respiration is calculated by subtracting OCR with rotenone/antimycin A from OCR at baseline. ATP-linked respiration was defined by the difference in OCR after the addition of oligomycin. Maximal respiration was calculated by subtracting OCR with rotenone/antimycin A from OCR with FCCP. Basal ECAR and Glycolytic capacity were defined by subtracting ECAR with 2-DG from ECAR at baseline and ECAR with oligomycin, respectively. All experiments were performed in triplicate using fresh preparations (*n* = 5). Data are presented as the mean ± the standard error of the mean (SEM). * *p* < 0.05 (ANOVA followed by a Tukey’s multiple comparison test).

**Figure 4 bioengineering-09-00327-f004:**
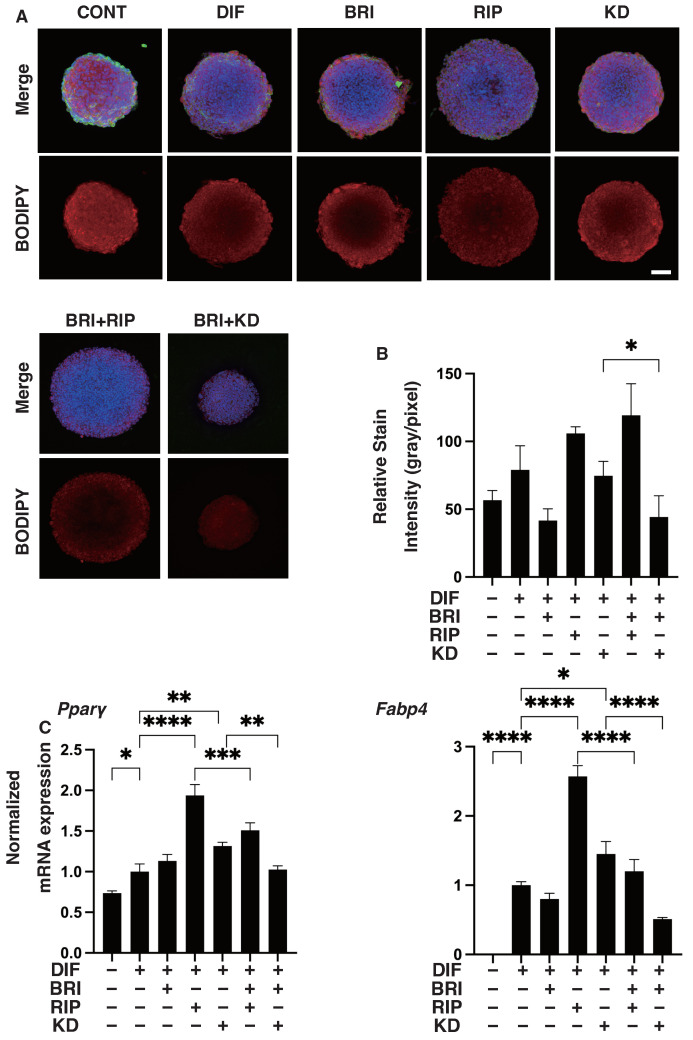
Effects of ROCK-i and/or BRI on adipogenic differentiation (DIF+) in 3D 3T3-L1 spheroids. The 3T3-L1 cells were 3D-cultured under several conditions: preadipocytes of 3T3-L1 cells (DIF−) and their adipogenic differentiation (DIF+) with or without 10 µM ripasudil (Rip) or KD025, and/or BRI. These were stained with BODIPY (**A**, scale bar: 100 μm), and their staining intensities were plotted (**B**). The mRNA expressions of DIF+-related genes, including Pparγ and Fabp4, under the above conditions were plotted in (**C**) 1–3. All experiments were performed in triplicate using fresh preparations, each consisting of five samples. Data are presented as the arithmetic mean ± standard error of the mean (SEM). * *p* < 0.05, ** *p* < 0.01, *** *p* < 0.005, **** *p* < 0.001, and other pairs were not statistically significant (ANOVA followed by a Tukey’s multiple comparison test).

**Figure 5 bioengineering-09-00327-f005:**
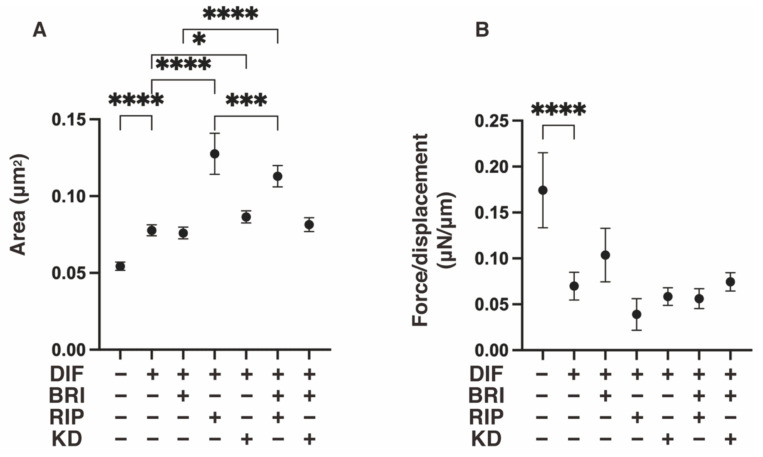
Effects of ROCK-i and/or BRI on the physical properties, size and stiffness of the 3T3-L1 3D spheroids. The 3T3-L1 cells were 3D-cultured under several conditions: preadipocytes of 3T3-L1 cells (DIF−) and their adipogenic differentiation (DIF+) with or without 10 µM ripasudil (Rip) or KD025, and/or BRI. The mean area sizes (μm^2^) were measured, and the results are plotted in panel (**A**). Alternatively, the stiffness was also measured using a microsqueezer, and the force required to induce deformation until half-diameter by 20 s (μN/μm force/displacement) was measured and is plotted in panel (**B**). All experiments were performed in triplicate using fresh preparations, each consisting of 16 spheroids. Data are presented as the arithmetic mean ± standard error of the mean (SEM). * *p* < 0.05, *** *p* < 0.005, **** *p* < 0.001, and other pairs were not statistically significant (ANOVA followed by a Tukey’s multiple comparison test).

**Figure 6 bioengineering-09-00327-f006:**
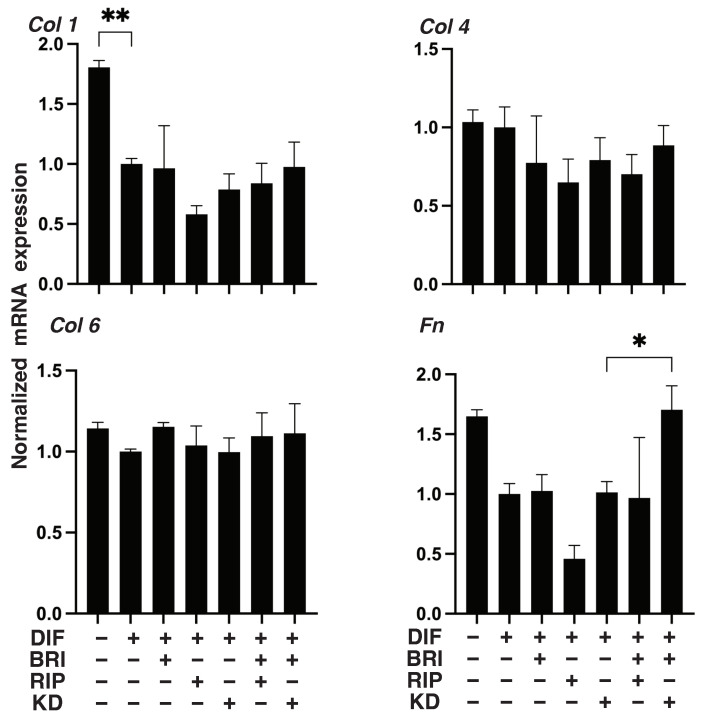
Effects of ROCK-i and/or BRI on the mRNA expression of ECMs in 3T3-L1 3D spheroids. The 3T3-L1 cells were 3D-cultured under several conditions: preadipocytes of 3T3-L1 cells (DIF−) and their adipogenic differentiation (DIF+) with or without 10 µM ripasudil (Rip) or KD025, and/or BRI were subjected to qPCR analysis to estimate the mRNA expression of ECM milecules (Col1: collagen 1, Col4: collagen 4, Col6: collagen 6, Fn: fibronectin). All experiments were performed in triplicate using fresh preparations, each consisting of five samples. Data are presented as the arithmetic mean  ±  standard error of the mean (SEM). * *p* < 0.05, ** *p* < 0.01, and other pairs were not statistically significant (ANOVA followed by a Tukey’s multiple comparison test).

## Data Availability

Not applicable.

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
