# Peer review of "Brimonidine Modulates the ROCK1 Signaling Effects on Adipogenic Differentiation in 2D and 3D 3T3-L1 Cells"

_bioengineering, 2022, doi:10.3390/bioengineering9070327_

Round 1

Reviewer 1 Report

The authors have presented a work to show that brimonidine effects adipogenesis in 2D and 3D models. Although the preliminary data are confirming to the story, the article is lacking sound experiments. I have the following query to the authors.

1.      The first sentence of the abstract is very long and complex, abstract is aimed at general reader, and it would be difficult to understand, therefore break that down into simple sentences.

2.    While the gene expression data seem convincing the general staining with BODIPY seems to be staining the cellular membrane, Can you specifically use some antibodies to show the adipogenic differentiation.

3.      Can you also consider showing protein data or immunofluorescence with specific antibody to prove the quantification of ECM proteins in both 2D and 3D, and draw the differences?

4.      Further use of IHC will enhance the results provided here.

5.      The methods to measure the intensity using Image J needs to be detailed

Author Response

The authors have presented a work to show that brimonidine effects adipogenesis in 2D and 3D models. Although the preliminary data are confirming to the story, the article is lacking sound experiments. I have the following query to the authors.

  1. The first sentence of the abstract is very long and complex, abstract is aimed at general reader, and it would be difficult to understand, therefore break that down into simple sentences.

Answer; Thank you for this comment. As suggested, the first sentence was changed into two sentences; “The additive effects of a a2-adrenergic agonist, brimonidine (BRI) on the pan-ROCK inhibitor (ROCK-i), Ripasudil (Rip) and the ROCK2-I, KD025 on adipogenic differentiation (DIF+) were examined using two- or three-dimension (2D or 3D) cultures of 3T3-L1 cells. The following analyses were carried out; 1) lipid staining (2D and 3D), 2) real time measurements of cellular metabolism (2D), 3) the mRNA expression of DIF+ related genes and extracellular matrix molecules (ECMs) including collagen (Col) -1, -4 and -6, and fibronectin (Fn) and 4) the sizes and physical properties of the 3D spheroids.”

  1. While the gene expression data seem convincing the general staining with BODIPY seems to be staining the cellular membrane, Can you specifically use some antibodies to show the adipogenic differentiation.

Answer; Thank you for this comment. Confirming adipogenic differentiation efficacies using antibodies to show the adipogenic differentiation or other methodology are clearly important issues. However, although as pointed out that BODIPY detected adipogenesis-related staining in addition to cellular membrane staining, the BODIPY method is still useful for estimating adipogenesis because, in our earlier study, we carefully evaluated several adipogenesis-related genes including Pparγ, Cebpa, Adipo Q Leptin, FABP4 and Glut 4 during the course of the maturation of adipogenesis in the 3D spheroids, and those adipogenesis efficacies were quite well correlated with the BODIPY staining results (ref; Sci Rep. 2020 May 14;10(1):7958., Sci Rep. 2021 Mar 9;11(1):5479.).  

  1. Can you also consider showing protein data or immunofluorescence with specific antibody to prove the quantification of ECM proteins in both 2D and 3D, and draw the differences?
  2. Further use of IHC will enhance the results provided here.

Answers for comments 3 and 4; Thank you for these comments. We agree that it will be much better to add additional data to support the qPCR analysis of ECM proteins. Therefore, immunofluorescein labeling of selected ECM proteins in which difference were observed among experimental conditions are now included in supplemental new figures 1 and 2.

  1. The methods to measure the intensity using Image J needs to be detailed

Answer; Thank you for this comment. As suggested, the use of macro for Image J is included in the supplemental table 2.

Reviewer 2 Report

 This research introduces DUES, a process in which PGs inhibit adipogenesis in orbital adipocytes, by mentioning that PGs are frequently used as first-line anti-glaucoma treatment, and explains the background well. In order to check whether other drugs that can be used as injections can affect adipogenesis in orbital adipocytes, a drug screening system was developed and drug efficacy against adipocytes was evaluated.

 To investigate the role of ROCK2 and α2-adrenergic signaling in 3T3-L1 cells, drug-induced effects were investigated by combining with pan-ROCK-i, Rip, specific ROCK2-i, KD025, α2-adrenergic agonist, brimonidine, BRI. The presentation of this part is very logical.

 In addition, as a result of these studies, this paper presents that α2-adrenergic stimulation of adipocytes induces the extracellular release of LPA resulting and α1-adrenergic vasoconstriction is linked to S1P2–G12/13–ROCK-mediated signaling. These are considered that this part is appropriate to enter into the discussion. Also, it is considered that the further study presented to overcome the limitations of this study is very desirable.

 However, the following should be corrected.

1.       In the Materials and Methods section, such as Seahorse XFe96 Bioanalyzer (Agilent Technologies), mention of country and city names is required.

2.       It seems that the correlation between the content of adipogenesis in orbital adipocytes and the content of 3T3-L1 mentioned in the discussion is insufficient.

3.       It seems necessary in the introduction section on how the inhibition of orbital adipocyte fat production affects the body.

4.       In the discussion, it is necessary to supplement the explanation that human adipocytes express all α2-adrenergic receptors of fat, whereas functional α2-Ars is not expressed in Rat.

5.       In Figure 1, the reason for not creating an experimental group of RIP + KD seems to be necessary.

6.       When presenting the same graph as the overall normalized mRNA expression, it would be better to present the fold value with the control set to 1.

7.       The integer part of μm2 of the figure legand part of Figure 4 should be indicated with superscript.

8.       It should indicate which word DIF stands for.

9.       It seems that the discussion part will be easier to understand when the mechanism and factors related to adipogenesis of 3T3-L1 are explained in the Introduction.

Author Response

This research introduces DUES, a process in which PGs inhibit adipogenesis in orbital adipocytes, by mentioning that PGs are frequently used as first-line anti-glaucoma treatment, and explains the background well. In order to check whether other drugs that can be used as injections can affect adipogenesis in orbital adipocytes, a drug screening system was developed and drug efficacy against adipocytes was evaluated.

 To investigate the role of ROCK2 and α2-adrenergic signaling in 3T3-L1 cells, drug-induced effects were investigated by combining with pan-ROCK-i, Rip, specific ROCK2-i, KD025, α2-adrenergic agonist, brimonidine, BRI. The presentation of this part is very logical.

 In addition, as a result of these studies, this paper presents that α2-adrenergic stimulation of adipocytes induces the extracellular release of LPA resulting and α1-adrenergic vasoconstriction is linked to S1P2–G12/13–ROCK-mediated signaling. These are considered that this part is appropriate to enter into the discussion. Also, it is considered that the further study presented to overcome the limitations of this study is very desirable.

 However, the following should be corrected.

  1. In the Materials and Methods section, such as Seahorse XFe96 Bioanalyzer (Agilent Technologies), mention of country and city names is required.

Answer; Thank you for this comment. As suggested, the country and city names are included; “a Seahorse XFe96 Bioanalyzer (Agilent Technologies, Santa Clara U.S.A.)”

  1. It seems that the correlation between the content of adipogenesis in orbital adipocytes and the content of 3T3-L1 mentioned in the discussion is insufficient.
  2. It seems necessary in the introduction section on how the inhibition of orbital adipocyte fat production affects the body.

Answers for questions 2 and 3; Thank you for these interesting questions. Almost no information has been available in terms of the differences and similarities between orbital adipocytes and other systemic adipocytes such as subcutaneous, visceral and brown adipocytes, and, to address this deficiency, our research group is currently working on this, although they should have different characteristics because of the fact that, in obesity, general adipocytes are increased but orbital adipocytes are not, but in turn, orbital adipocytes but not general adipocytes are increased in patients with Graves’ disease. Therefore, we do not have precise information concerning the correlation between the magnitude of and other aspects of adipogenesis in orbital adipocytes and the content of 3T3-L1 cells, as well as another question as to how inhibiting orbital adipocyte fat production affects the body. Therefore, this information is included in the study limitations in the discussion section; “However, there as limitations to this study, the following issues will need to be considered; First, the roles of ROCK1 in the process of DIF+ are not fully understood since, among the ROCKs, ROCK2 but not ROCK1, is recognized as being mainly responsible for the anti-adipogenic activity toward 3T3-L1 cells [51]. Therefore, additional investigations with the objective of elucidating the roles of ROCK1 and 2 signaling within the DIF+ as well as their relationship with a2-adrenergic signaling will need to be done. This will require the use of additional experimental groups such as Rip and KD025, and other pan-ROCK-is molecules with different suppressive efficacies with respect to ROCK1 and 2. Second, in the current study, we used 3T3-L1 cells, a mouse embryonic fibroblast cell line to study the drug induced effects toward orbital fatty tissues. However, these adipocytes may have different characteristics, since it is known that obesity is associated with an increase in general adipocytes but not orbital adipocytes [52], but in turn, orbital adipocytes but not general adipocytes are increased in patients with Graves’ disease [53]. Third, the drug-induced effects of the mRNA expressions of ECMs molecules were quite different between 2D and 3D cell cultures although the lipid staining and mRNA expression of DIF+ related genes were similar. As a possible explanation for this difference, we speculate that DIF+ may prefer a 3D spatial space rather than the 2D plane space. In fact, in a previous study, we identified the spontaneous adipogenic differentiation of the 3D 3T3-L1 preadipocytes without their induction, although such spontaneous DIF+ was not detected in the 2D cultured cells. To identify possible mechanisms responsible for causing such difference between 2D and 3D cell culture, RNA sequence analyses were performed. The results suggested that STAT3 functions as the master regulator in forming the 3D spheroid architecture, and related signaling may induce these unique biological differences [54]. However, as of his writing, the relationship between DIF+ and STAT3-related signaling as well as diversity in the natures of orbital and other systemic adipocytes have not been fully investigated. Therefore, additional study using several specific inhibitors, SiRNA knockdown and others strategies will also be required to reveal the relationship between biological environments, e.g., 2D vs 3D, and several biological signaling networks using several sources of adipocytes, i.e., both orbital and other systemic adipocytes.”.  

  1. In the discussion, it is necessary to supplement the explanation that human adipocytes express all α2-adrenergic receptors of fat, whereas functional α2-Ars is not expressed in Rat.

Answer; Thank you for this comment. As suggested, additional references related to functional α2-Ars within human adipocytes are now cited; “However, such adrenergic regulation of lipolysis is known to be exclusively species dependent, that is, human fat cells express both lipolytic β- and antilipolytic α2-adrenergic receptors (α2-ARs) [42,43], whereas no functional α2-ARs are found in rat fat cells [41]. In fact, an antilipolytic a2-adrenergic effect is well known within human adipocytes [44,45].”.

  1. In Figure 1, the reason for not creating an experimental group of RIP + KD seems to be necessary.

Answer; Thank you for this comment. We agree with such an additional experimental group of RIP and KD, and other ROCK-is conditions with different inhibitions with respect to ROCK1 and 2 would be advisable to get further scientific insights into understanding the roles of ROCK 1 and 2. Therefore, this information is included within the study limitation within the last paragraph of the Discussion section; “However, there as limitations to this study, the following issues will need to be considered; First, the roles of ROCK1 in the process of DIF+ are not fully understood since, among the ROCKs, ROCK2 but not ROCK1, is recognized as being mainly responsible for the anti-adipogenic activity toward 3T3-L1 cells [51]. Therefore, additional investigations with the objective of elucidating the roles of ROCK1 and 2 signaling within the DIF+ as well as their relationship with a2-adrenergic signaling will need to be done. This will require the use of additional experimental groups such as Rip and KD025, and other pan-ROCK-is molecules with different suppressive efficacies with respect to ROCK1 and 2. Second, in the current study, we used 3T3-L1 cells, a mouse embryonic fibroblast cell line to study the drug induced effects toward orbital fatty tissues. However, these adipocytes may have different characteristics, since it is known that obesity is associated with an increase in general adipocytes but not orbital adipocytes [52], but in turn, orbital adipocytes but not general adipocytes are increased in patients with Graves’ disease [53]. Third, the drug-induced effects of the mRNA expressions of ECMs molecules were quite different between 2D and 3D cell cultures although the lipid staining and mRNA expression of DIF+ related genes were similar. As a possible explanation for this difference, we speculate that DIF+ may prefer a 3D spatial space rather than the 2D plane space. In fact, in a previous study, we identified the spontaneous adipogenic differentiation of the 3D 3T3-L1 preadipocytes without their induction, although such spontaneous DIF+ was not detected in the 2D cultured cells. To identify possible mechanisms responsible for causing such difference between 2D and 3D cell culture, RNA sequence analyses were performed. The results suggested that STAT3 functions as the master regulator in forming the 3D spheroid architecture, and related signaling may induce these unique biological differences [54]. However, as of his writing, the relationship between DIF+ and STAT3-related signaling as well as diversity in the natures of orbital and other systemic adipocytes have not been fully investigated. Therefore, additional study using several specific inhibitors, SiRNA knockdown and others strategies will also be required to reveal the relationship between biological environments, e.g., 2D vs 3D, and several biological signaling networks using several sources of adipocytes, i.e., both orbital and other systemic adipocytes.”.

  1. When presenting the same graph as the overall normalized mRNA expression, it would be better to present the fold value with the control set to 1.

Answer; Thank you for this comment. As suggested, the qPCR graphs are now presented the fold value with the control set to 1.

  1. The integer part of μm2 of the figure legand part of Figure 4 should be indicated with superscript.

Answer; Thank you for this comment. As pointed out, the corresponding sentence was changed; “The mean area sizes (μm2)” in Fig. 5 legend.

  1. It should indicate which word DIF stands for.

Answer; Thank you for this comment. As suggested, to understand this easier, we used preadipocyte (DIF-) and adipogenic differentiation (DIF+).

  1. It seems that the discussion part will be easier to understand when the mechanism and factors related to adipogenesis of 3T3-L1 are explained in the Introduction.

Answer; Thank you for this comment. As suggested, the mechanism responsible for adipogenesis and its regulation are included within 3rd paragraph of the introduction; “Adipocytes have various pivotal functions and properties including lipid storage, energy homeostasis, and insulin sensitivity, and their adipogenic differentiation (DIF+) are critically regulated by transcriptional factors including peroxisome proliferator-activated receptor γ (PPARγ) as well as others [15] and are influenced by several signaling molecules including PGs [16], ROCK and others [17]. Recently, we also investigated the effects of ROCK inhibitors (ROCK-is) on DIF+ using 2D and 3D cultured 3T3-L1 cells, since ROCK-is are known to reduce the deposits of ECM proteins within the trabecular meshwork (TM), resulting in IOP reduction in several animal models [18,19], and one pan-ROCK-i, ripasudil hydrochloride hydrate (Rip) is already approved for the treatment of glaucoma and ocular hypertension [20,21]. Interestingly, the results revealed that the addition of ROCK-i beneficially abrogated the PG induced suppression of DIF+ in 3T3-L1 cells, suggesting that ROCK-i may be one of the preferred choices as a second line glaucoma medication for reducing the DUES that is induced by PGs [22]”.

Round 2

Reviewer 1 Report

The article can now be accepted